# Information Clues and Emotional Intentions: A Case Study of the Regional Image of the Cultural and Creative Community

Yulin Chen

Department of Mass Communication, Tamkang University, New Taipei City 25137, Taiwan; 143530@mail.tku.edu.tw

**Abstract:** In order to capture the value of cultural creativity, this study explored regional cultural creativities with different creative forms to understand how people interpret and interact with various regional cultural creative images. This was done by analyzing the abstract (performance) type of cultural creativity and the figurative (commodity) type of cultural creativity, in order to understand how existing regional cultural creativities operate information threads in social media, and how the different forms of content may lead to different levels of participation and feedback. The Cloud Gate Dance Theater can be taken as an example of an abstract cultural creation (performance type), and Green-in-hand as an example of a figurative cultural creativity (commodity type). In this study, all user page content for the period 1 January 2011, to 31 December 2018, and the number of user comments for each post were analyzed, for a total of 4784 posts. Computer science, data mining, big data, and social network analysis were combined to verify the findings of the analyses. Through an application programming interface (API), data and information in social media is extracted. Then data filtering, storage, and analysis is performed with meaningful information extracted for interpretation and for use in text mining to explore the relationship with the public based on content attributes. This study first verifies that the regional image is consistent with the social image location. Second, the information cues results found that information cues could be organized into region personality through any direct or indirect contact. Third, emotional clues can evoke emotions and self-expression, which is seen as an important clue to region emotions. In addition, this study also provides a conceptual framework for understanding how different forms of information, in regards to social management of existing regional cultural creativities, leads to varying levels of participation. Understanding the form of information is a key factor in the acceptance of information by the public. It is a reminder for cultural and creative institutions of the importance of text and images, and of figurative and abstract information planning in social content. In order to improve the competitiveness of the destination, using content interaction through social media to create and enhance a strong brand image is important.

**Keywords:** social media content exploration; regional image; abstract and figurative clues; cultural and creative community

---

## 1. Introduction

Culture uses experiences to convey meaning (Dewey 1934), which can be achieved through demonstrations or everyday experiences (Newman et al. 2012). Culture may come from any part of life with the experience is shared, such as by common symbols, texts, or information, by any group, thus conveying beliefs, customs, values, and other figurative or abstract practices that confirm the cultural common identity (Throsby 2001). Cultural creativity can be regarded as a combination of experience

and art. It is a kind of understanding of the meaning of culture to individuals or people, such as the recognition and interpretation of information, and the meaning of symbols (Throsby 2001). The latter may be integrated into the arts and create a dialogue within the culture, just as individuals participate in cultural creation while sharing experiences (McCarthy et al. 2001). It also represents the value of cultural creativity in people's lives (Throsby 2001). Therefore, in order to capture the value of cultural creativity, this study explored regional cultural creativities with different creative forms to understand how people interpret and interact with various regional images of cultural creativity (Gray 2010). This study also explored how subjective thinking produces a regional image after being influenced by the text and images (Tribe and Xiao 2011). However, as the composition of the regional image is very complex, Gallarza et al. (2002) considered it to be a complex, multi-dimensional, and dynamic combination that cannot be explained by a simple facet (Gallarza et al. 2002). Therefore, the regional image of cultural creativity discussed in this study is an example of images and words in social content. By subjectively judging the masses in order to generate behavioral interactions between emotions and intentions, it produces a specific figurative or abstract impression (Baloglu and McCleary 1999). In order to conceptualize the regional image (Canally 2010), this study applies the concept of converting a vague regional image into a concrete symbol (Therkelsen 2003). This is an issue that is highly valued in many image studies. For example, assessing the individual attributes of the regional image and obtaining specific factors that influence the regional image (Jenkins 1999). The common definition of the regional image refers to the potential ideal image in the minds of the masses (Mak 2011). Information obtained through television, the internet, books, and magazines can be transformed into figurative or abstract concepts (Gartner 1994; Jenkins 1999). Moreover, information from different media is often used to guide the masses in a systematic manner (Ekinci 2003). Many ideas and concepts induced by secondary images have caused prejudice before the actual contact with the region (Gartner 1994). Therefore, in order to explore the individual's information perception, it is necessary to analyze the mass perception of the region image and the primary and secondary image relationships in the minds of individuals which can help to adjust the correct positioning of the region to the masses (Fesenmaier and MacKay 1996).

In view of this, this study refers to the above literature, by analyzing the abstract (performance) type of cultural creativity, and the figurative (commodity) type of cultural creativity, in order to understand how existing regional cultural creativities operate information threads in social media, and how the different forms of content may lead to different levels of participation and feedback. In addition, this study focuses on the relationship between content participation, information cues, and emotional cues, which attempt to amplify information issues in social business strategies. When the text content is different from the image content, there may be negative emotions and behaviors. This study will discuss the social content marketing perspective in regards to cultural creativity, and examine the image positioning and actual information benefits of cultural creativity. Taking the Cloud Gate Dance Theater as an example of an abstract cultural creation (performance type), and Green-in-hand as an example of a figurative cultural creativity (commodity type), the research presented herein combines abstract cultural creativity and figurative cultural creativity to interpret these two completely different regional cultural creativities. This study examines the characteristics of constructing text and image content, and seeks to grasp the characteristics or problems of the current regional cultural creativity's information management from the actual behavior data. It will be investigated if there will be a gap in thinking for social content in the public because of their own tonal differences. It is hoped that the results of this study can be used as a reference for future cultural creations or related cultural institutions in social marketing.

There are three research purposes for this study. First, this study is different from other business types of region research, as it simply targets non-profit regional cultural creativities as the object, using social media content exploration technology to collect post content, mass behavior, and emotional data for analysis. By comparing figurative cultural creativity and abstract cultural creativity how region socialization succeeds in achieving complex cultural and creative ideas and images through clear image

positioning can be understood. Secondly, this study explores the relationship between information, mass emotions, and behavior using a clear behavioral model. While existing research enriches our understanding of mass behavior (Kohler et al. 2011; Nambisan and Baron 2007), in the face of changing social media, this study uses social information (text, photo, video) and mass emotions ('Love', 'Haha', 'Wow', 'Sorry', 'Anger'), correlated with variables, such as behaviors (likes, comments, shares), to explain the relationship between social information and mass participation. Thirdly, an attempt is made to find poor information positioning and determine how to transform the content, enhance the public's perception and emotions, and enhance public participation in order to achieve a more effective social dialogue. The framework developed in this study will effectively examine the motivations and behavioral responses of the masses to social media participation in cultural institutions.

The second section of this study briefly describes the relevant literature and theories of regional image, information clues, and emotional clues at the present stage of research on this topic. Section 3 proposes the cultural creative powder specialization, from the perspective of text and image content, to explore relevant assumptions between content and mass participation, and uses this to assess the impact of information clues on the mood and behavior of the masses. Section 4 focuses on the methodology of the research process, with Section 5 presents the data analysis. Section 6 discusses the results, suggesting how to use this model to carry out the content management of the relevant cultural creative social media and develop the ideal content marketing plan.

## 2. Theoretical Background

### 2.1. Regional Image Analysis with Content Exploration

The regional image can effectively drive the perception of the masses (Guthrie and Gale 1991), representing the impressions and concepts of the masses on the target (Baloglu and McCleary 1999). Since the image can express the objective psychology of the masses (Myers 1968), Crompton (1979) suggested that using images to guide images is the most effective method (Crompton 1979). Embacher and Buttle (1989) also pointed out that the regional image represents an individuals' personal thoughts and impressions (Embacher and Buttle 1989), so the image is not only transformed into the cognition of the masses, but also can be influenced by various means (Baloglu and McCleary 1999; Chon 1990; Gallarza et al. 2002; Prayag and Ryan 2012). Beerli and Martín (2004) extends the regional image to represent individuals, groups, knowledge, impressions, prejudice, and emotional performance of specific goals, by combining cultural and social attributes (Beerli and Martín 2004), or combining history, politics, economics, and other different aspects that have an impact (Gartner 1994). Therefore, the constituent elements of the regional image may mainly include social and cultural factors, projection factors, media factors, and mass-generated content factors (Gartner 1994; Jenkins 1999). These factors do have an irreplaceable influence on the formation of the regional image. Images that represent the regional image, whether figurative or abstract, provide hints of the mass audience's more targeted experience. The content delivered will also have an impact. For example, formal and informal content, communicated through the media or spoken language, will indirectly affect the comments and reactions of potential visitors.

After consolidating the current literature on the regional image, it is found that discussion has not been limited to the regional image concept (Gallarza et al. 2002), the regional image composition (Baloglu and McCleary 1999), and the regional image influence (Bigné et al. 2001; Echtner and Ritchie 1993). Significant research has been performed on the decision-making of the regional image to the masses (Heitmann 2011), and the survey of the differences in outcomes caused by the regional image (Goodrich 1978; Hosany et al. 2006). Further research on the online text of the regional image found that most of the research focused on the calculation and measurement of the regional image (Choi et al. 2007; Govers et al. 2007; Pan 2011; Stepchenkova and Morrison 2006), the test of regional image theory (Papathanassis and Knolle 2011), as well as regional image case analysis and measurement, etc. (Li et al. 2015). Most of the regional image online text measurement

research used word processing software to process the online text information of the regional image, and then determined the relationship between various word frequencies and regional image. In the field of cognitive psychology, many studies have attempted to distinguish between cognitive and emotional associations in different types of images (Burns et al. 1993). Images trigger multiple sensory perceptions, such as smell, hearing, touch, and taste, so images and human visual perceptions have complex interactions.

Among them, the presentation of the content of the regional image is more likely to attract the attention of the masses, and directly generate the perception and impression of the image (Mackay and Fesenmaier 1997). Since images can freely transform personal experiences and shape the unique image of the target (Chalfen 1979), images can be used to construct symbolic effects in memory by shaping the perception and imagination of the target (Haldrup and Larsen 2003) and providing a direct sensory response in the audience (Hum et al. 2011). In particular, images in social media can increase the familiarity and trust of specific targets for potential people (Trauer and Ryan 2005). It can be explained that images are like providing an ideology (Liesch 2011), so a visual analysis of online regional images will help managers grasp the respective dimensions of the image, such as text or image dimensions, abstract and figurative dimensions, etc. (Kim and Stepchenkova 2015; Pan et al. 2014). At the same time, this is combined with social media sharing, viewing, and message features (Vu et al. 2015), so as to promote content sharing (Lo et al. 2011; Stepchenkova and Zhan 2013), or social information analysis of image communication (Lo et al. 2011). Therefore, in addition to text, it is also important to include region image-assisted content presentation (Hancock and Toma 2009). Using symbolic representations of the content, image visualization can be performed to enhance the value of images in region dialogue (Hunter 2015). Matteucci used images to explore the perception of the regional image (Matteucci 2013), while Pennington compared online images generated by marketers of the same region to verify the need for image content for the regional image (Pennington and Thomsen 2010).

Unfortunately, there is currently little research on content targeting and regional images (Gallarza et al. 2002). Considering that today's information content is becoming more and more extensive, there is still a lack of research on the effect of multiple sources of information and different media types. Therefore, this study specifically refers to the influence research of the above regional image (Baloglu and McCleary 1999; Echtner and Ritchie 1991), and related research on social media, which rethinks the relationship between the regional image and social content (Baloglu and McCleary 1999; Pike 2002). Through content analysis applications, the regional image in text and images is more widely explored, and comprehensive measurement of text, image, or multimedia information is achieved.

## 2.2. Task Clues and Emotional Clues in Information

The frequency of interaction between the masses and regional cultural creatives in social media often involves the reconciliation of information positioning (Animesh et al. 2011). The so-called reconciliation refers to whether the masses have the willingness to participate actively in social media, interpreting the information and indirectly reflecting the degree of region recognition of the masses (Jiang et al. 2010). Eroglu et al. suggested that common information cues can be divided into task cues and emotional cues, mainly by examining the level of demand for social cues (Eroglu et al. 2001; Parboteeah et al. 2009; Zhang 2013). This is because the degree of reference for the public is mainly influenced by task clues and emotional clues (Wang and Zhang 2012). Task clues represent the usefulness of information and the time the people take to process the information and make decisions (Parboteeah et al. 2009). Eroglu et al. verified the importance of information cognition for feedback, and the results confirmed that high cue clues and low cue clues have different effects on mass purchasing behavior (Eroglu et al. 2001). Parboteeah suggested that information cognition could assess the appropriateness of information presentation, such as the degree of visual or functional attraction. It was also found through experimental results that high-quality information clues could effectively increase the possibility of online impulse purchases (Parboteeah et al. 2009). In recent years, there have been many image studies turning to social media to collect data and explore the value of images in

social media (Mariani et al. 2016; Molinillo et al. 2018). For example, Munar and Jacobsen obtained data from TripAdvisor and Flickr to explore the interaction between time structure, communication scope, number of social threads and content richness (Munar and Jacobsen 2014). Clore et al.'s assessment of region awareness and sentiment found that the way people deal with social information affects the attitude and evaluation of subsequent messages (Clore et al. 2001; Schwarz et al. 1991).

Information cues are often used to enhance information cognition. For example, video may be used due to its strong visual attraction and relative ease in improving information understanding and response because of the high degree of information integrity (Loiacono et al. 2007). Therefore, the cognition and emotional response of the masses to visual elements can be used as a reference for judging whether the information meets the task (Eroglu et al. 2001). Language or content clues can reflect the degree of mastery of information (Zhang 2013), while emotional clues tend to be emotional in information (Zhang 2013). For example, music on video sites (Wu et al. 2008), images, or animated content, all have unique emotional clues (Chowdhury et al. 2008; Park et al. 2008), or elements that affect emotional processing (Zhang 2013), in order to make viewers have a pleasant experience (Eroglu et al. 2001). Visual elements, images, and videos enhance information performance and actively stimulate public sentiment (Parboteeah et al. 2009). Moreover, results show that information clues and emotional clues are complementary to each other. In addition to satisfying the information transmission, it is desirable to create a sense of happiness for viewers.

At present, there are many surveys and analyses related to content and emotions on the topic of social media. Most of the methods transform the original data into useful information and knowledge, and then interpret the data with appropriate techniques and methods (Abrahams et al. 2013). Among them, the classification of unstructured data is especially suitable for social media research. For example, Chen et al. suggest that keywords and high-frequency words are strong indicators for social media topics (Chen et al. 2013). Classification techniques for different topics also provide an unprecedented advantage for behavioral prediction (Chen and Liu 2004). Researchers and analysts can find with the optimal social media action by discovering research on social content issues (Barbier and Liu 2011). For example, He et al. used text mining to analyze the content and unstructured information of Facebook and Twitter in the three major US pizzerias, comparing the differences between the two social media platforms (He et al. 2013), and collecting social media data and data cleansing for text mining and content analysis. Key topics or classifications were explored and extracted from which important classification paths between data was found (He et al. 2013). At the same time, through the comparison and application of two different social media platforms, the content of the social media information was understood (He et al. 2013). Referring to the above literature, this study attempts to further expand the discussion of information clues and emotional clues. First, different types of regional cultural creativities are focused on, with an analysis of how their social information influences the masses. Then through the masses' perceptions, emotions, and behaviors, an information benefit evaluation is carried out with details, such as the presentation form and information frequency. Finally, the classification information is used to propose the cause of the behavior gap due to the difference between the text and the image content.

## 3. Research Model and Hypothesis

### 3.1. Textual Imagery and Graphic Imagery in the Regional Image

The regional image has a considerable role in individuals' decision-making process (Baloglu and McCleary 1999; Beerli and Martín 2004), which directly affects the follow-up assessment and behavioral intentions of individuals. Current research defines the characteristics of the regional image in various ways. Gallarza et al. define the impression, perception, and feeling of the masses on the target (Gallarza et al. 2002; Zhang et al. 2014), which includes different levels of cognition, knowledge, thought, and emotion (Baloglu and McCleary 1999; Beerli and Martín 2004; Chew and Jahari 2014). Baloglu et al. assessed the effects of cognitive and emotional interactions by assessing cognition, emotion,

and intention (Baloglu and McCleary 1999; Tseng et al. 2015). This model is also widely used as the regional image theory framework for follow-up research. The theoretical framework mainly evaluates the relationship between cognitive impression and emotional impression (Chew and Jahari 2014). Cognitive impression refers to the cognition and knowledge about the target, which is related to the characteristics of the target. For example, the theme or symbol is based on the cognitive response of the regional image (Wang and Hsu 2010), while the emotional impression refers to the emotions and feelings felt by the public about the target, and the intention represents the future actions or intentions. These actions may involve different behavioral responses such as comments and actual participation (Gartner 1994). Gartner further proposes a hierarchical causal model, in which cognition affects emotions and intentions, and emotion also affects the depth of cognition. Cognitive and emotional components are interrelated and largely depend on the informational influence of cognitive sources (Baloglu and McCleary 1999), and the type or form of information is an important reason for guiding cognition (Beerli and Martín 2004; Li et al. 2010; Stern and Krakover 1993).

In addition, in research on regional images and emotional responses, Lee et al. particularly emphasized the perceived value in the regional image (Lee et al. 2005), especially for satisfaction (Prayag 2009) and the will to visit (the region). Intentions can have an impact (Chew and Jahari 2014). Tseng et al. further divided the composition of the regional image into three phases (Tseng et al. 2015). The first phase is mainly to induce and modify the induced image stage, which tends to be the basic construction of cognitive impressions (Gunn 1988). The second stage is the cognition during the perception by the public, which is transformed into the constructive emotional stage and intention in the emotional impression. This becomes the common theoretical framework of the follow-up regional image. In the third stage, the regional image is directly linked to the image positioning, and the interaction between the two is reviewed (Pike 2002). In reference to the above-mentioned theory of emotion and behavior, it is evident that the presentation of information and the impact of targeting on individuals are important, and regional cultural creativities are more focused on what kind of text or image content to set. Therefore, this study focuses on the cultural and creative art of the figurative (commodity) and the abstract (performance) cultural and creative art regional cultural creativities. The analysis of the text and image content of the region community may have different emotional impact and behavioral participation of the masses (Figure 1). The hypotheses are as follows:

**Hypothesis 1 (H1).** *Regional cultural creativities use the regional image for packaging, and its link post (LP, pure text) has an impact on the emotional and behavioral participation of the social masses.*

**Hypothesis 1a (H1a).** *Performance culture and its link post (LP) have an impact on emotional social participation.*

**Hypothesis 1b (H1b).** *Performance culture and its link post (LP) have an impact on the social participants' behavioral participation.*

**Hypothesis 1c (H1c).** *Merchandise culture and its link post (LP) have an impact on the emotional participation of social people.*

**Hypothesis 1d (H1d).** *Merchandise culture and creativity and its link post (LP) have an impact on the social participants' behavioral participation.*

**Hypothesis 2 (H2).** *Regional cultural creativities use the regional image for packaging, and their photo posts (PP, image and text) have an impact on the emotional and behavioral participation of social masses.*

**Hypothesis 2a (H2a).** *Performance culture creative photo posts (PP) have an impact on the emotional participation of social masses.*

**Hypothesis 2b (H2b).** *Performance culture and its photo posts (PP) have an impact on the social participants' behavioral participation.*

**Hypothesis 2c (H2c).** *Merchandise culture creative photo posts (PP) have an impact on the emotional participation of social masses.*

**Hypothesis 2d (H2d).** *Merchandise culture and creativity and its photo posts (PP) have an impact on the social participants' behavioral participation.*

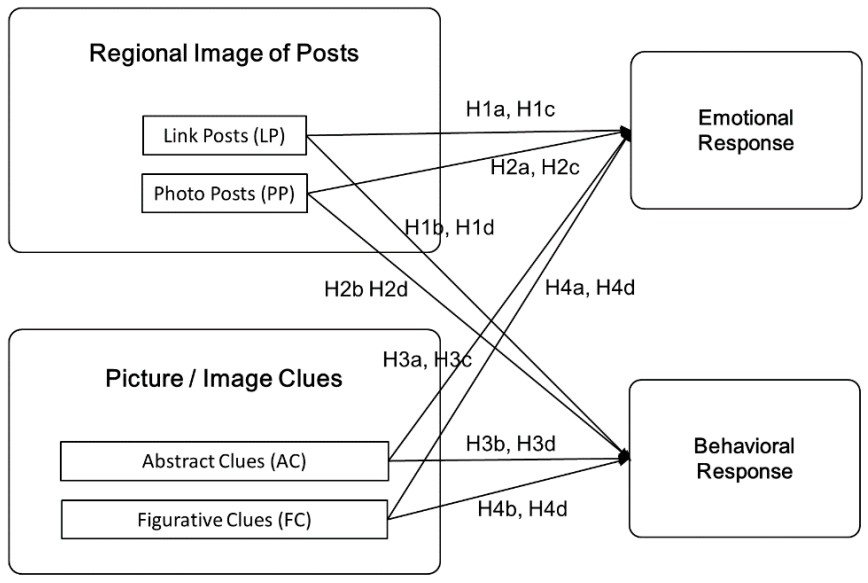

**Figure 1.** Extended research model.

*3.2. Abstract Clues and Figurative Clues in Image Information*

According to the current research related to information, social postings have different information characteristics. For example, Rauschnabel et al. (2012) classify social content with different characteristics, such as text frequency, media elements (such as images), and perception surveys (Rauschnabel et al. 2012). De Vries et al. used six common functions in addition to content, distinguishing the possibility of features that guide reading, such as posting on the page, as well as using the reaction after reading. These investigators suggested that interactive, information rich, vivid, and interesting content is critical (De Vries et al. 2012). In addition, based on the psychological model (Hansen 1976), the posting can be classified into different forms such as cognitive information, emotional information, and service information (De Vries et al. 2012).

Regardless of the type of information, its image will undoubtedly affect reactions to the image (Mehrabian and Russell 1974). From prior studies discussing e-commerce, it is evident that information has a positive impact on the perception of the masses (Parboteeah et al. 2009). The diversified information is easy to satisfy public region curiosity, and may even provide greater entertainment for people (Muniz and O'Guinn 2001). Of course, if the integrity of the information is higher, the probability of achieving the task will be relatively higher, and the emotional response of the group will also increase

(Nambisan and Baron 2007). For example, Animesh et al. have confirmed that abstract information helps to construct entertainment value, and the more clearly information is positioned, the easier it is for people to have a pleasant experience (Animesh et al. 2011; Schau et al. 2009). Therefore, after reference to the information and reaction-related literature (Heijden et al. 2003; Parboteeah et al. 2009), this study presumes that image information can indeed drive people to generate emotional or behavioral responses. However, the focus needs to be on the assessment of informational content, i.e., whether there is a difference in participation due to the representation or abstraction of the presentation. The classification proposed in this study is based on using image analysis technology for the visual elements in the content, exploring the concrete task clues and abstract emotional clues, and the differences in the emotions and intentions of the recipients. The hypotheses are as follows:

**Hypothesis 3 (H3).** *The information content of the regional cultural creativity presents images of abstract (feeling) clues, which have an impact on the emotional and behavioral participation of social masses.*

**Hypothesis 3a (H3a).** *The content of performance culture and creativity presents images of abstract (feeling) clues, which have an impact on the emotional participation of social masses.*

**Hypothesis 3b (H3b).** *The content of performance culture and creativity presents images of abstract (feeling) clues, which have an impact on the behavioral participation of social masses.*

**Hypothesis 3c (H3c).** *The content of merchandise culture and creativity presents images of abstract (feeling) clues, which have an impact on the emotional participation of social masses.*

**Hypothesis 3d (H3d).** *The content of merchandise culture and creativity presents images of abstract (feeling) clues, which have an impact on the behavioral participation of social masses.*

**Hypothesis 4 (H4).** *The information content of the regional cultural creativity presents images of figurative (task) clues, which have an impact on the emotional and behavioral participation of social masses.*

**Hypothesis 4a (H4a).** *The content of performance culture and creativity presents images of figurative (task) clues, which have an impact on the emotional participation of social masses.*

**Hypothesis 4b (H4b).** *The content of performance culture and creativity presents images of figurative (task) clues, which have an impact on the behavioral participation of social masses.*

**Hypothesis 4c (H4c).** *The content of merchandise culture and creativity presents images of figurative (task) clues, which have an impact on the emotional participation of social masses.*

**Hypothesis 4d (H4d).** *The content of merchandise culture and creativity presents images of figurative (task) clues, which have an impact on the behavioral participation of social masse.*

## 4. Research Methodology

Social Media Content Exploration (SMCM) refers to collecting social media platform content, text, or data (He et al. 2013), including text mining, to explore models or trend relationships. SMCM refers to using content mining to capture images, videos, relevant information, or data, such as multimedia, and discussing data processing of non-text structures (Kaplan and Haenlein 2010). An example is using Facebook tools and technology to collect and filter social media content (Fayyad et al. 1996; Liu and Park 2015). The development of social media has continually led to an in-depth analysis of content exploration (Zeng and Gerritsen 2014). First, through an application programming interface (API), data and information in social media are extracted. Then data filtering, storage, and analysis is

performed with meaningful information extracted for interpretation based on the analysis of data results (Filieri and McLeay 2014). The SMCM application provides a variety of social content characteristics recognition patterns that understand the causes and meanings of human behavior in terms of functions, information, and themes. Qualitative and quantitative analysis is carried out as a basis for judging the trend of unstructured content. Social media data can be collected in an instant, thus allowing managers to react quickly and relocate resources. SMCM uses tools, such as collection, exploration, and visualization, to detect and analyze social media data (Tang and Yang 2017). Regardless of the use of manual modeling or computational techniques for semi-automatic modeling, SMCM is one of the most common methods (Chanana et al. 2004). Therefore, it is often used in theoretical research on social content or through text mining to explore the relationship with the public based on content attributes (He et al. 2013; Chandrasekaran et al. 1999).

The implementation of social media content exploration in this study is described below. First, the application program (API) is used to extract data from social media, then information and data collation, integration, and storage are executed. Finally, the image analysis technology "Jumptuit" of Google Cloud Vision is used to automatically detect the image elements, such as characters and objects in images or films, which includes optical character recognition, interactive label detection, document text detection, face detection, web detection, and logo detection. Then, these data are automatically classified and integrated in the database through machine learning. Moreover, the application of artificial intelligence technology, combined with image analysis and machine learning technology, can analyze the facial expressions and emotions of the portrait according to the category of search. In order to integrate the needs of region image research, web detection and logo detection are especially adopted to analyze the collected community images, and then filter and integrate the important analysis results.

The Cloud Gate Dance Theater and Green-in-hand are two of the few cultural and creative institutions authorized and promoted by the Executive Yuan and the Ministry of Culture. Therefore, its brand image and popularity are highly recognized by the government and the public. According to our observations, this institution is also more active in community content planning and interaction with the audience. Therefore, given that it is easier to obtain sufficient data sources for this institution, samples from the Cloud Gate Dance Theater and Green-in-Hand were selected for analysis in this study. Previously, the issue considered in this study was tested with a 2018 posts. However, it was found that the test results were not significant due to the small number of posts, so it was decided to increase the time range of the posts under consideration. Since 2011, the tested samples began to have a relatively stable frequency of postings. Therefore, it was decided to collect all the post materials from 2011 to 2018 on a yearly basis. Using a stable post frequency and the largest number of fans, a social media content analysis was performed for both Taiwan's Cloud Gate Dance Theater and Green-in-hand, two different regional cultural creativities. We collected all user page content for the period 1 January 2011 to 31 December 2018, and the number of user comments for each post. There were a total of 4784 posts. Of the posts, 1299 were links, and 3485 had photo content. Of the interactions, there were 921,487 Likes, 11,861 Comments, and 69,661 Shares. Of the emotional responses, there were 7558 for Love, 674 for Haha, 1080 for Wow, 583 for Sorry, and 20 for Anger (Table 1).

**Table 1.** Total number of posts.

| | Post (N) | Likes | Comments | Shares | Behavioral Participation | Love | Haha | Wow | Sorry | Anger | Emotional Participation |
|---|---|---|---|---|---|---|---|---|---|---|---|
| **Artist positioning brands (AP)** | | | | | | | | | | | |
| Link post | 910 | 39,428 | 1098 | 3498 | 44,934 | 365 | 83 | 177 | 22 | 9 | 656 |
| Photo post | 4052 | 343,965 | 7802 | 22,632 | 378,451 | 3161 | 1312 | 1323 | 95 | 25 | 5916 |
| **Ordinary people positioning brands (OP)** | | | | | | | | | | | |
| Link post | 738 | 198,371 | 3706 | 14,011 | 216,826 | 1105 | 431 | 1078 | 81 | 31 | 2726 |
| Photo post | 5476 | 2,195,549 | 27,449 | 68,587 | 2,297,061 | 5856 | 4137 | 3033 | 620 | 193 | 13,839 |
| Sum | 11,176 | 2,777,313 | 40,055 | 108,728 | 2,937,272 | 10,487 | 5963 | 5611 | 818 | 258 | 23,137 |

## 5. Data Analyses and Results

### 5.1. Reliability and Validity

In this study, factor analysis was performed to assess the reliability and effectiveness of the data. The KMO value obtained in this study was 0.741, with Cronbach's α being found as 0.722. The factor load is close to or higher than 0.7, indicating good convergence and discriminant validity. Two statistical analyses, namely simple correlation analysis and linear regression analysis, were performed in this study. In addition, this study conducted multiple tests to examine the correlation between independent variables. A variance inflation factor (VIF) exceeding 10 indicates multiple collinearity problems. In this study, the value of VIF is exclusively lower than 10, indicating that there was no multicollinearity.

### 5.2. Hypothesis Testing

First, we tested Hypothesis 1, that regional cultural creativities use the regional image for packaging, and its link post (LP, pure text) (Table A1), has an impact on the emotional and behavioral participation of the social masses. Of these, H1a, performance culture and its link post (LP), and H1c, merchandise culture and its link post (LP), having impacts with the emotional participation of social media users, are partially supported. H1b, performance culture and its link post (LP) having an impact on the social participants' behavioral participation, is also partially supported. However, H1d, merchandise culture and creativity and its link post (LP) having an impact on the social participants' behavioral participation, is marginally supported.

Second, we tested Hypothesis 2, that regional cultural creativities use the regional image for packaging, and their photo posts (PP, image and text) have an impact on the emotional and behavioral participation of social masses. It was found that H2b, performance culture creative photo posts (PP), and H2d, merchandise culture and creativity and its photo posts (PP), having impacts with the behavioral participation of social media users, are supported. H2a, performance culture creative photo posts (PP) having an impact on the emotional participation of social masses, is also supported. However, H2c, merchandise culture creative photo posts (PP) having an impact on the emotional participation of social masses, is partially supported

Next, we tested Hypothesis 3, that the information content of the regional cultural creativity presents images of abstract (feeling) clues, which have an impact on the emotional and behavioral participation of social masses. The results show that H3a, the content of performance culture and creativity presents images of abstract (feeling) clues, and H3c, the content of merchandise culture and creativity presents images of abstract (feeling) clues, having impacts with the emotional participation of social media users, are both marginally supported. H3b, the content of performance culture and creativity presents images of abstract (feeling) clues, and H3d, the content of merchandise culture and creativity presents images of abstract (feeling) clues, all having impacts with the emotional participation of social media users, are supported.

Finally, we tested Hypothesis 4 that the information content of the regional cultural creativity presents images of figurative (task) clues, which have an impact on the emotional and behavioral participation of social masses. H4a, the content of performance culture and creativity presents images of figurative (task) clues, which have an impact on the emotional participation of social masses, is partially supported. H4b, the content of performance culture and creativity presents images of figurative (task) clues, which have an impact on the behavioral participation of social masses, is supported. However, H4c, the content of merchandise culture and creativity presents images of figurative (task) clues, which have an impact on the emotional participation of social masses, and H4d, the content of merchandise culture and creativity presents images of figurative (task) clues, which have an impact on the behavioral participation of social masses, are both marginally supported (Table 2).

**Table 2.** Summary of findings.

| ID | Hypothesis | Verdict |
|---|---|---|
| H1. | The link posts (LP, pure text) have an impact on the emotional and behavioral participation of people. | |
| H1a. | The link posts (LP) of "artist" positioning have an impact on the emotional participation of people. | Marginally supported |
| H1b. | The link posts (LP) of "artist" positioning have an impact on the behavioral participation of people. | Marginally supported |
| H1c. | The link posts (LP) of "ordinary people" positioning have an impact on the emotional participation of people. | Partial supported |
| H1d. | The link posts (LP) of "ordinary people" positioning have an impact on the behavioral participation of people. | Partial supported |
| H2. | The photo posts (PP, image and text) have an impact on the emotional and behavioral participation of people. | |
| H2a. | The photo posts (PP) of "artist" positioning have an impact on the emotional participation of people. | Supported |
| H2b. | The photo posts (PP) of "artist" positioning have an impact on the behavioral participation of people. | Supported |
| H2c. | The photo posts (PP) of "ordinary people" positioning have an impact on the emotional participation of people. | Supported |
| H2d. | The photo posts (PP) of "ordinary people" positioning have an impact on the behavioral participation of people. | Supported |
| H3. | The abstract implication (emotional) pictures have an impact on the emotional and behavioral participation of people. | |
| H3a. | The abstract implication (emotional) pictures of "artist" positioning have an impact on the emotional participation of people. | Marginally supported |
| H3b. | The abstract implication (emotional) pictures of "artist" positioning have an impact on the behavioral participation of people. | Supported |
| H3c. | The abstract implication (emotional) pictures of "ordinary people" positioning have an impact on the emotional participation of people. | Partial supported |
| H3d. | The abstract implication (emotional) pictures of "ordinary people" positioning have an impact on the behavioral participation of people. | Supported |
| H4. | The concrete implication (missionary) pictures have an impact on the emotional and behavioral participation of people. | |
| H4a. | The concrete implication (missionary) pictures of "artist" positioning have an impact on the emotional participation of people. | Supported |
| H4b. | The concrete implication (missionary) pictures of "artist" positioning have an impact on the behavioral participation of people. | Supported |
| H4c. | The concrete implication (missionary) pictures of "ordinary people" positioning have an impact on the emotional participation of people. | Partial supported |
| H4d. | The concrete implication (missionary) pictures of "ordinary people" positioning have an impact on the behavioral participation of people. | Supported |

## 6. Discussion and Implications

### 6.1. Discussion of Findings

This research produced some interesting findings. The results show that the planning of the contents of the social media pages of the regional cultural creativities significantly affected the users' emotional responses to the content and behavioral participation (Figure 2).

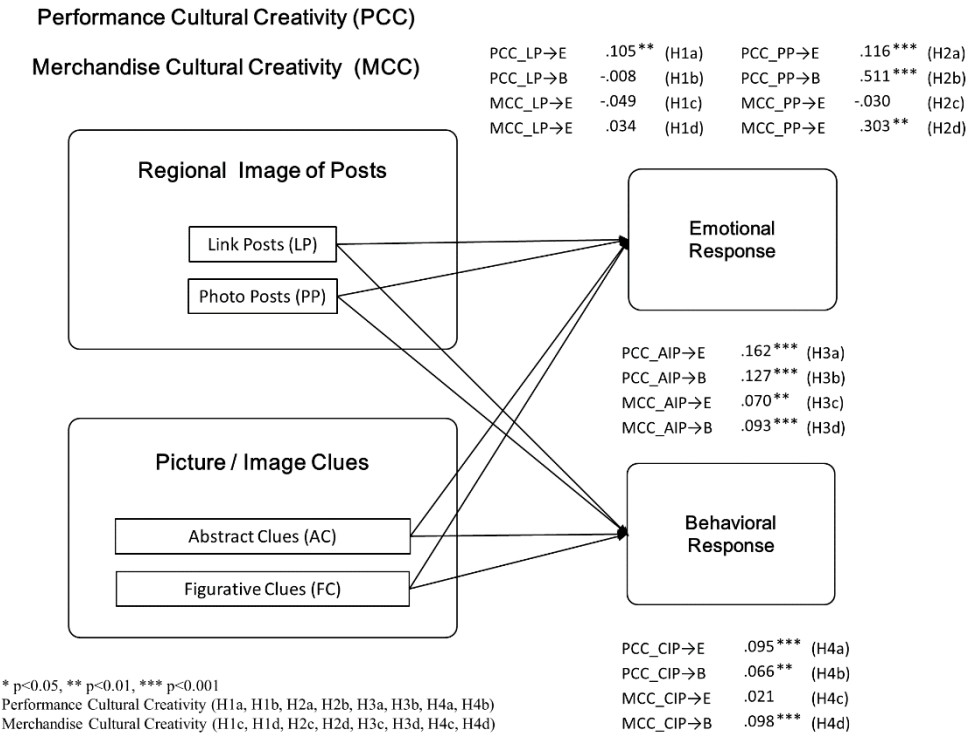

**Figure 2.** Model results.

First, we tested Hypothesis 1, that the image of regional cultural creativities with link posts (LP, pure text), have an impact on the emotional and behavioral participation of people. For the emotional participation of users, the performance culture and its link post (LP) had values of β = 0.104, *t* = 3.097, and *p* < 0.001, and the behavioral participation of users had values of β = −0.008, t = −0.242. For the emotional participation of users, the merchandise culture and its link post the LP had values of β = −0.049, t = −1.022, and the behavioral participation of users had values of β = 0.034, t = 0.707. All three regional cultural creativities have statistically significant results.

Second, we tested Hypothesis 2, that the image of regional cultural creativities with photo posts (PP, image and text), have an impact on the emotional and behavioral participation of people. For the emotional participation of users, the performance culture creative photo posts (PP) had values of β = 0.115, t = 4.810, and *p* < 0.001, and the behavioral participation of users had values of β = −0.155, t = −6.487, and *p* < 0.001. For the emotional participation of users, the merchandise culture creative photo posts (PP) had values of β = −6.487, t = −1.268, and the behavioral participation of users had values of β = −0.079, t = −3.359, and *p* < 0.005. All three regional cultural creativities have statistically significant results.

Next, we tested Hypothesis 3, that the image of regional cultural creativities with the abstract implication (emotional) pictures, have an impact on the emotional and behavioral participation of people. For the emotional participation of users, the content of performance culture and creativity presents images of abstract (feeling) clues had values of β = 0.162, t = 6.765, and *p* < 0.001, and the behavioral participation of users had values of β = 0.127, t = 5.291, and *p* < 0.001. For the emotional participation of users, the content of merchandise culture and creativity presents images of abstract (feeling) clues had values of β = 0.069, t = 2.962, and *p* < 0.005, and the behavioral participation of users had values of β = 0.092, t = 3.925, and *p* < 0.001. All three regional cultural creativities have statistically significant results.

Finally, we tested Hypothesis 4, that the image of regional cultural creativities with the concrete implication (missionary) pictures, have an impact on the emotional and behavioral participation of people. In terms of the emotional participation of users, the content of performance culture and creativity presents images of figurative (task) clues had values of β = 0.094, t = 3.914, and *p* < 0.001, and

the behavioral participation of users had values of β = 0.066, t = 2.740, and *p* < 0.01. For the emotional participation of users, the content of merchandise culture and creativity presents images of figurative (task) clues had values of β = 0.020, t = 0.878, and the behavioral participation of users had values of β = 0.097, t = 4.154, and *p* < 0.001.

*6.2. Theoretical Implications and Limitations and Future Research*

The results of this study found that, first of all, pictures with text could elicit better emotional and behavioral responses (Tables A2 and A3). Especially for abstract cultural ideas of expression classes, image-type posts have a significant relationship in terms of emotions. The figurative cultural creativity of goods, using images, can produce significant results in behavioral responses, such as message or sharing. Therefore, in regards to the information clues and emotional reactions, we find that the most common emotion caused by cultural creative powder is Love, followed by Haha and Wow. In particular, the emotional response of the graphic form will be greater than the simple text content. The reason is that the information in graphic form is the most direct and the most informative, so it is the easiest to stimulate the understanding and input of the masses (Keller 1993).

Secondly, whether it is performance culture or commodity culture, if the image content contains a smile, there will be obvious significant results in the interactive results of the sharing. Further examination of the emotional response reveals that performance-based cultural creativity has results that are more significant for Wow and Sorry in terms of emotions, while commodity-based cultural creativity tends to favor positive emotions, such as Love and Haha. This shows that mastering the characteristics of two completely different regional cultural creativities can indeed lead the public to respond more effectively to the corresponding emotions. Information clues are the benchmarks for the masses to judge the image. Emotional clues are opinions or likes and dislikes that the masses have on the image based on the content received. The masses interpret different regional cultural creativities (Netemeyer et al. 2004) for informational experience, such as region knowledge, region feedback and region impressions, etc. (D. A. Aaker 1996).

Finally, this study analyzed the abstract and figurative imagery of the image, and found that the performance culture has achieved remarkable results due to its abstract image and the region name or performance image in the emotional clue (Tables A4–A6). Commodity culture and creativity are also based on the characteristics of the region. The more obvious the region symbol for the image clues presented by the information clues, the easier it is to have dialogue with the public, resulting in feedback behavior, such as Likes and Shares, as well as a performance culture. Therefore, understanding the region characteristics and presenting the figurative or abstract representation of the image is the key to content marketing. When the public experiences social media, they immediately respond to the acceptance of the social information, and reflect their satisfaction in emotions and behaviors (Jiang et al. 2010). Therefore, consistent with previous studies, it is confirmed that entertainment is a key factor influencing social behavior (Lin and Lu 2011; Sledgianowski and Kulviwat 2009). This is because information rich in entertainment elements is more likely to trigger positive comments from the masses and even stimulate a higher likelihood of individuals' revisiting (Raney et al. 2003).

This study suggests that the graphical information and social content management of regional cultural creativities can be evaluated for the following three points.

First, regardless of any type of region, it is important for text to match images for social content, so that informational clues can effectively match emotional clues and improve the efficiency of information dissemination.

Second, the regional cultural creativity of the performance type can use graphic and textual communication for emotional clues, and the cultural creativity of the commodity type can enhance the figurative region or product information according to its own figurative characteristics, so that the visual and region image are consistent. This enhances communication skills.

Third, if you want to enhance the expression of human nature, you can try to add a portrait or a smiling face. According to the image analysis results of this study, it can be seen that the presentation

of smiley symbols has a significant auxiliary effect on the positive emotions of the masses and the willingness to share content.

*6.3. Academic Contribution/Practical Contribution*

On the academic level, this study has the following contributions.

This study first verifies that the regional image is consistent with the social image location. The regional image reflects the degree of influence and informational experience (Kent and Allen 1994). The key is whether information associations can be evoked in individuals' memory and become the interpretation adopted by the public. Familiar information is clearly easier to recall than unfamiliar. Therefore, the information-familiar and visually relevant hypothesis model has been specifically demonstrated in advertising images and memory recall studies (Campbell and Keller 2003). Whether it is from information organization theory or information processing-related research, it has been repeatedly shown that information familiarity is highly correlated with memory recall (Campbell and Keller 2003). The complete and clear content of the clues can preserve more powerful region memories and make it easier to activate memories for region links (Kent and Kellaris 2001).

Second, the information cues results found that information cues could be organized into region personalities through any direct or indirect contact (Plummer 1985). It is generated through different marketing programs, such as product-related attributes, prices, advertisements, etc. (Batra et al. 1993), and further becomes a region identity (Fournier 1998). Of course, information can also be used to motivate the masses to act (Fournier 1998). Cognition not only represents the beliefs and ideas of the masses, but also represents their emotional responses (Bagozzi 1978). In general, emotional or behavioral responses are often seen as the basis for information location (Srull and Wyer 1989) and can be effectively used as the primary analytical tool for social content.

Third, emotional clues can evoke emotions and self-expression (Donahay and Rosenberger 2007), which are seen as important clues to region emotions (Freling and Forbes 2005). The emotional identity of the region can be used to check the integrity of the image information in the regional image (Baloglu and McCleary 1999), for example, whether the figurative or abstract elements in the image can be clearly conveyed. The emotional identity is a key factor that may also affect the subjective formation.

In addition, two primary contributions result from this study.

First, this study validates the importance of information cue and emotional cue models for region images, and changes with figurative and abstract perceptions (Keller 2001). Social media provides marketers with the opportunity to increase region exposure, and it is necessary to strengthen the region image to enhance the region knowledge of the masses. Regional cultural creativities and people communicate with each other without time, place, or media restrictions, forming a two-way interaction between the public and regional cultural creativities (Vargo and Lusch 2008). Communication cannot only enhance image and mass relationships (Kim and Ko 2012), the active participation of the public will also affect the content participation and perceived value by the public (Christodoulides et al. 2012). Therefore, frequent information experience and communication, region associations and attitudes towards the public will have a positive impact (D. A. Aaker 1991). In particular, based on image information, social media activities are continuously constructed to reduce misunderstandings and prejudice against regional cultural creativities and, thus, enhance positive value in regional cultural creativities through information exchange (Kim and Ko 2012).

Second, this study used experiments to measure the public after receiving the information, to assess the interaction between the public and different regional cultural creativities, to examine why the media information is out of balance with the needs of the public, and to verify the importance of using informational clues to influence public participation (J. L. Aaker 1997). It is recommended that managers stimulate the public to generate region identity and participation by strengthening information with emotional content. Moreover, in terms of the special type of fan page data calculation, the enhanced behavioral response of Comment and Share will be better than Likes. The efficient use of information leads to effective interactions, raising people's message and sharing behaviors, and stimulating the

further reach of content (Hoyer et al. 2010). In addition, this study also provides a conceptual framework for understanding how different forms of information, in regards to social management of existing regional cultural creativities, leads to varying levels of participation. Understanding the form of information is a key factor in the acceptance of information by the public. It is a reminder for cultural and creative institutions of the importance of text and images, and figurative and abstract information planning in social content.

### 6.4. Research Limitations and Recommendations

A primary research limitation lies in the classification method of information clues. The method used only uses Facebook's existing link form posts and image type posts, and the analysis results are inevitably limited. In the future, we expect relevant research to be able to target more detailed categories, such as announcement-type information, promotional information, or interactive information, etc., to determine what information the region should provide in order to more easily respond to today's changing needs.

Secondly, in view of the application of images in the cultural and creative industries, it must be considered that the culture is composed of many complex beliefs, values, and ideas, resulting in differences in attitudes and approaches. Therefore, communication strategies and region statements in cultural creativity should be chosen as a communication tool that easily extends the impression and creates benign communication (Harvey 1993; Mueller 1992). It is also encouraged to attempt to analyze image-specific image positioning (Sriram and Gopalakrishna 1991), combining cultural creative features with social image content exploration as an important operational tool (Dunn 1976). According to different media characteristics, different levels of element cutting are better for the information needs of the public (Green et al. 1975; Pollay 1986).

**Funding:** This research received no external funding.

**Acknowledgments:** This study was funded by the [Ministry of Science and Technology—Digital Humanities Program] grant number [0610234].

**Conflicts of Interest:** The author declares no conflict of interest. The author declares that: (i) no support, financial or otherwise, has been received from any organization that may have an interest in the submitted work; and (ii) there are no other relationships or activities that could appear to have influenced the submitted work.

## Appendix A. Measurement and Items

**Table A1.** Total number of text frequencies and implication frequencies.

| Artist Positioning Brands (AP) | | | | | | Ordinary People Positioning Brands (OP) | | | | | |
|---|---|---|---|---|---|---|---|---|---|---|---|
| Text Frequencies of Photo Post | N | Text Frequencies of Link Post | N | Implication Frequencies of Picture | N | Text Frequencies of Photo Post | N | Text Frequencies of Link Post | N | IMPLICATION Frequencies of Picture | N |
| Huashan | 6428 | Huashan | 1747 | design | 1828 | Barge two | 4708 | Barge two | 642 | design | 2735 |
| Http | 3131 | Http | 1244 | product | 976 | art | 4312 | art | 546 | art | 2579 |
| activity | 2855 | activity | 795 | art | 847 | warehouse | 3467 | Kaohsiung | 476 | product | 1379 |
| time | 2067 | Goo.gl | 510 | Creative | 719 | design | 2947 | design | 461 | font | 981 |
| Kyrgyzstan | 2003 | Wenchuang | 481 | park | 667 | Kaohsiung | 2498 | Http | 390 | center | 829 |
| Report time | 2001 | Kyrgyzstan | 462 | Huashan | 576 | Http | 2397 | we | 275 | Pier2 | 760 |
| art | 1492 | Report time | 461 | font | 543 | artist | 1826 | warehouse | 228 | Graphic | 708 |
| life | 1491 | time | 451 | tree | 538 | exhibition | 1748 | everyone | 183 | Brand | 526 |
| Show | 1314 | Taiwan | 435 | illustrator | 448 | we | 1700 | artist | 180 | sky | 518 |
| Park | 1269 | art | 426 | Graphic | 402 | time | 1691 | Erbo | 176 | illustrator | 504 |
| we | 1208 | design | 401 | house | 389 | Big meaning | 1570 | can | 175 | tree | 455 |
| Taiwan | 1191 | Park | 380 | sky | 379 | Admission | 1426 | exhibition | 173 | exhibition | 438 |
| location | 1159 | Sign up | 346 | Brand | 348 | Erbo | 1395 | time | 157 | computer | 415 |
| everyone | 1151 | life | 346 | advertising | 295 | Two art | 1270 | the film | 142 | street | 412 |
| exhibition | 1144 | location | 343 | Street | 282 | Barge two art | 1268 | together | 138 | Kaohsiung | 398 |
| Goo.gl | 1128 | we | 339 | Public | 258 | Refuting art | 1267 | Design section | 133 | city | 378 |
| Wenchuang | 1084 | share it | 306 | computer | 255 | Goo.gl | 1250 | youth | 132 | poster | 360 |
| design | 1051 | Creative | 301 | music | 248 | activity | 1227 | Goo.gl | 130 | line | 347 |
| together | 943 | everyone | 287 | line | 204 | everyone | 1196 | Barge two art | 127 | Public | 306 |
| Exhibition | 893 | Brand | 285 | poster | 198 | location | 1175 | activity | 126 | group | 306 |
| on site | 889 | can | 280 | Roof | 187 | wristband | 1063 | Refuting art | 126 | advertising | 305 |
| Sign up | 881 | exhibition | 277 | Performing | 180 | creation | 1045 | Special zone | 120 | photography | 299 |
| date | 819 | More | 248 | Tourism | 175 | works | 1013 | Sign up | 115 | music | 290 |
| friend | 786 | jobs | 243 | area | 171 | can | 963 | creation | 112 | museum | 285 |
| creation | 767 | One | 242 | Human | 167 | passport | 962 | Art district | 112 | culture | 282 |
| Experience | 765 | Own | 242 | Service | 165 | Passport hand | 959 | Taiwan | 109 | Logo | 260 |
| limited | 743 | Exhibition | 241 | square | 163 | Passport bracelet | 958 | location | 108 | Tourism | 258 |
| can | 730 | creation | 240 | Facade | 159 | Da Yong | 927 | Creative | 107 | film | 258 |
| Brand | 715 | culture | 234 | architecture | 158 | Special zone | 883 | jobs | 106 | light | 255 |
| Creative | 691 | limited | 234 | Interior | 156 | jobs | 869 | works | 105 | Pier2 | 250 |
| culture | 690 | Creative park | 230 | photography | 154 | Special | 863 | Admission | 105 | Yellow | 247 |
| Market | 685 | Wenchuang Park | 229 | space | 151 | Artistic | 862 | Refuting the Second Art District | 104 | square | 228 |
| One | 676 | together | 221 | cartoon | 149 | together | 853 | Big meaning | 98 | Photograph | 227 |
| course | 666 | Reading | 218 | flower | 147 | Special zone | 848 | open | 98 | artist | 226 |
| Own | 662 | illustration | 205 | Exhibit | 146 | Art district | 847 | News | 90 | paint | 221 |
| One day | 652 | Reading Huashan | 204 | Image | 146 | Refuting Art | 843 | More | 87 | Wallpaper | 218 |
| story | 640 | Read Huashan | 202 | taipei | 144 | Refuting the Second Art District | 842 | One | 85 | Angle | 213 |
| jobs | 629 | Show | 199 | Facebook | 138 | Passport hand | 826 | Kaohsiung City | 85 | area | 213 |
| works | 611 | Taipei | 195 | Photograph | 137 | Passport bracelet | 825 | course | 83 | Human | 211 |
| Rihuashan | 602 | story | 190 | culture | 136 | the University | 802 | photography | 82 | Desktop | 209 |

**Table A1.** *Cont.*

| | | | | | | | | | | | |
|---|---|---|---|---|---|---|---|---|---|---|---|
| **Artist Positioning Brands (AP)** | | | | | | **Ordinary People Positioning Brands (OP)** | | | | | |
| **Text Frequencies of Photo Post** | **N** | **Text Frequencies of Link Post** | **N** | **Implication Frequencies of Picture** | **N** | **Text Frequencies of Photo Post** | **N** | **Text Frequencies of Link Post** | **N** | **IMPLICATION Frequencies of Picture** | **N** |
| One day Huashan | 593 | invite | 190 | text | 133 | Taiwan | 799 | Ticket sales | 77 | Text | 206 |
| period | 588 | Market | 181 | Wallpaper | 132 | life | 740 | culture | 76 | Bureau | 202 |
| Id | 585 | date | 181 | group | 130 | Market | 719 | remember | 76 | Affairs | 193 |
| world | 583 | works | 177 | Desktop | 126 | Design section | 697 | Through | 76 | Government | 193 |
| music | 582 | Lecture | 173 | meter | 125 | Nowadays | 693 | space | 73 | Pattern | 190 |
| Taipei | 580 | Id | 166 | Pattern | 119 | Contemporary | 666 | Purchase tickets | 72 | pier-2 | 184 |
| Wenchuang Garden | 578 | friend | 165 | behavior | 116 | Penglai | 659 | what | 71 | meter | 183 |
| Wenchuang Park | 576 | Curation | 164 | dance | 116 | Resident in the village | 650 | share it | 71 | vehicle | 178 |
| News | 555 | workshop | 162 | Logo | 115 | the film | 649 | friend | 71 | Image | 177 |
| More | 536 | industry | 162 | food | 114 | One | 638 | wristband | 70 | service | 177 |

## Appendix B. The Results of Independent Sample *t* Test and Linear Regression

**Table A2.** The results of independent sample t test (artist positioning (AP)).

| | Huashan | | | | Wenchuang | | | | Art | | | | Design | | | | Park | | | |
|---|---|---|---|---|---|---|---|---|---|---|---|---|---|---|---|---|---|---|---|---|
| | F | *p*-Value | t-Value | *p*-Value | F | *p*-Value | t-Value | *p*-Value | F | *p*-Value | t-Value | *p*-Value | F | *p*-Value | t-Value | *p*-Value | F | *p*-Value | t-Value | *p*-Value |
| **Artist positioning (AP)_link post** | | | | | | | | | | | | | | | | | | | | |
| Likes | 2.334 | 0.127 | −0.704 | 0.482 | 4.325 | 0.038 | −1.136 | 0.256 | 1.369 | 0.242 | 0.504 | 0.615 | 0.033 | 0.856 | −0.558 | 0.577 | 1.027 | 0.311 | 0.020 | 0.984 |
| Comments | 0.092 | 0.762 | −0.199 | 0.842 | 2.744 | 0.098 | −0.974 | 0.330 | 2.977 | 0.085 | 0.962 | 0.336 | 2.119 | 0.146 | −0.896 | 0.370 | 1.036 | 0.309 | −0.463 | 0.643 |
| Shares | 0.141 | 0.707 | −0.168 | 0.866 | 3.764 | 0.053 | −0.834 | 0.404 | 4.753 | 0.030 | 1.048 | 0.296 | 0.281 | 0.596 | −0.409 | 0.682 | 0.436 | 0.509 | 0.011 | 0.991 |
| Behavioral | 1.601 | 0.206 | −0.620 | 0.536 | 4.322 | 0.038 | −1.175 | 0.241 | 2.257 | 0.133 | 0.712 | 0.477 | 0.070 | 0.791 | −0.593 | 0.553 | 0.911 | 0.340 | −0.020 | 0.984 |
| Love | 0.004 | 0.947 | 0.224 | 0.823 | 0.040 | 0.842 | −0.008 | 0.993 | 0.677 | 0.411 | 0.652 | 0.514 | 0.600 | 0.439 | 0.707 | 0.480 | 1.547 | 0.214 | 0.879 | 0.380 |
| Haha | 5.477 | 0.019 | 1.941 | 0.053 | 3.495 | 0.062 | −0.954 | 0.340 | 2.424 | 0.120 | −0.784 | 0.433 | 0.895 | 0.344 | −0.491 | 0.623 | 2.678 | 0.102 | −0.830 | 0.407 |
| Wow | 0.706 | 0.401 | 0.453 | 0.651 | 8.613 | 0.003 | −1.936 | 0.053 | 2.159 | 0.142 | −0.720 | 0.472 | 0.162 | 0.688 | −0.219 | 0.827 | 0.088 | 0.767 | −0.092 | 0.927 |
| Sorry | 6.320 | 0.012 | −0.898 | 0.370 | 6.517 | 0.011 | −1.635 | 0.103 | 1.073 | 0.300 | 0.524 | 0.600 | 4.797 | 0.029 | −1.956 | 0.051 | 1.897 | 0.169 | −0.689 | 0.491 |
| Anger | 13.740 | 0.000 | −1.524 | 0.129 | 2.259 | 0.133 | −0.749 | 0.454 | 30.519 | 0.000 | 1.771 | 0.078 | 0.071 | 0.789 | 0.136 | 0.892 | 1.958 | 0.162 | −0.698 | 0.486 |
| Emotional | 0.315 | 0.574 | 0.566 | 0.571 | 5.344 | 0.021 | −1.522 | 0.128 | 0.415 | 0.520 | −0.181 | 0.857 | 0.110 | 0.741 | −0.049 | 0.961 | 0.220 | 0.639 | −0.016 | 0.987 |
| **Artist positioning (AP)_photo post** | | | | | | | | | | | | | | | | | | | | |
| Likes | 22.164 | 0.000 | −1.263 | 0.207 | 41.527 | 0.000 | −9.083 | 0.000 | 5.853 | 0.016 | −2.351 | 0.019 | 46.530 | 0.000 | −10.673 | 0.000 | 20.767 | 0.000 | −5.417 | 0.000 |
| Comments | 4.335 | 0.037 | −0.970 | 0.332 | 0.496 | 0.481 | 0.343 | 0.731 | 15.501 | 0.000 | 1.236 | 0.217 | 4.535 | 0.033 | −3.412 | 0.001 | 2.093 | 0.148 | 0.939 | 0.348 |
| Shares | 11.474 | 0.001 | −1.075 | 0.282 | 3.268 | 0.071 | −1.190 | 0.234 | 0.212 | 0.645 | −0.052 | 0.958 | 6.328 | 0.012 | −4.004 | 0.000 | 0.145 | 0.703 | −0.221 | 0.825 |
| Behavioral | 23.305 | 0.000 | −1.375 | 0.169 | 31.554 | 0.000 | −8.065 | 0.000 | 3.167 | 0.075 | −1.801 | 0.072 | 39.253 | 0.000 | −10.190 | 0.000 | 14.141 | 0.000 | −4.502 | 0.000 |
| Love | 0.503 | 0.478 | 0.813 | 0.416 | 2.470 | 0.116 | −0.953 | 0.341 | 1.849 | 0.174 | 0.422 | 0.673 | 4.573 | 0.033 | −2.097 | 0.036 | 0.526 | 0.468 | −0.413 | 0.680 |
| Haha | 1.087 | 0.297 | 0.541 | 0.589 | 12.332 | 0.000 | −3.192 | 0.001 | 6.167 | 0.013 | −1.694 | 0.090 | 7.976 | 0.005 | −2.544 | 0.011 | 9.071 | 0.003 | −2.283 | 0.023 |
| Wow | 1.199 | 0.273 | 0.678 | 0.498 | 3.241 | 0.072 | −1.022 | 0.307 | 3.276 | 0.070 | 0.835 | 0.404 | 3.940 | 0.047 | −2.660 | 0.008 | 0.438 | 0.508 | −0.326 | 0.745 |
| Sorry | 2.224 | 0.136 | −0.747 | 0.455 | 7.064 | 0.008 | −1.806 | 0.071 | 25.253 | 0.000 | −4.683 | 0.000 | 10.434 | 0.001 | −2.467 | 0.014 | 1.929 | 0.165 | −0.696 | 0.486 |
| Anger | 1.040 | 0.308 | −0.515 | 0.607 | 12.819 | 0.000 | 1.312 | 0.190 | 2.152 | 0.142 | −0.733 | 0.464 | 2.935 | 0.087 | 0.864 | 0.388 | 26.659 | 0.000 | 1.980 | 0.048 |
| Emotional | 0.626 | 0.429 | 0.796 | 0.426 | 6.532 | 0.011 | −2.465 | 0.014 | 0.513 | 0.474 | −0.025 | 0.980 | 7.002 | 0.008 | −3.158 | 0.002 | 2.550 | 0.110 | −0.871 | 0.384 |

**Table A3.** The results of independent sample t test (ordinary people positioning (OP)).

| | Pier2 | | | | Art | | | | Kaohsiung | | | | Design | | | | Exhibition | | | |
|---|---|---|---|---|---|---|---|---|---|---|---|---|---|---|---|---|---|---|---|---|
| | F | p-Value | t-Value | p-Value | F | p-Value | t-Value | p-Value | F | p-Value | t-Value | p-Value | F | p-Value | t-Value | p-Value | F | p-Value | t-Value | p-Value |
| **Ordinary people positioning (OP)_link post** | | | | | | | | | | | | | | | | | | | | |
| Likes | 0.087 | 0.768 | −0.196 | 0.845 | 6.221 | 0.013 | −2.186 | 0.029 | 2.218 | 0.137 | 1.395 | 0.163 | 12.334 | 0.000 | −4.484 | 0.000 | 8.163 | 0.004 | −3.741 | 0.000 |
| Comments | 0.921 | 0.338 | 0.284 | 0.776 | 8.614 | 0.003 | −2.624 | 0.009 | 0.775 | 0.379 | 0.679 | 0.497 | 19.438 | 0.000 | −5.086 | 0.000 | 1.466 | 0.226 | −1.138 | 0.256 |
| Shares | 2.438 | 0.119 | −0.473 | 0.636 | 19.453 | 0.000 | −3.512 | 0.000 | 4.740 | 0.030 | 1.169 | 0.245 | 10.778 | 0.001 | −3.505 | 0.000 | 6.500 | 0.011 | −3.159 | 0.002 |
| Behavioral | 0.015 | 0.902 | −0.222 | 0.825 | 7.655 | 0.006 | −2.380 | 0.018 | 2.402 | 0.122 | 1.420 | 0.156 | 13.020 | 0.000 | −4.502 | 0.000 | 8.121 | 0.004 | −3.746 | 0.000 |
| Love | 14.477 | 0.000 | 3.212 | 0.001 | 7.362 | 0.007 | −1.864 | 0.063 | 41.702 | 0.000 | −10.103 | 0.000 | 7.279 | 0.007 | −1.664 | 0.097 | 0.218 | 0.641 | 0.181 | 0.856 |
| Haha | 1.040 | 0.308 | 0.634 | 0.527 | 1.729 | 0.189 | 0.619 | 0.536 | 14.043 | 0.000 | −5.217 | 0.000 | 7.146 | 0.008 | −2.285 | 0.023 | 4.586 | 0.033 | 1.414 | 0.160 |
| Wow | 0.019 | 0.891 | 0.137 | 0.891 | 7.434 | 0.007 | −2.092 | 0.037 | 7.704 | 0.006 | −4.140 | 0.000 | 7.428 | 0.007 | −2.938 | 0.003 | 2.083 | 0.149 | −0.745 | 0.456 |
| Sorry | 2.547 | 0.111 | 0.816 | 0.415 | 3.163 | 0.076 | −0.898 | 0.370 | 3.459 | 0.063 | −0.940 | 0.348 | 5.360 | 0.021 | −2.286 | 0.023 | 1.807 | 0.179 | −0.683 | 0.495 |
| Anger | 2.162 | 0.142 | 0.735 | 0.463 | 10.783 | 0.001 | −2.434 | 0.015 | 4.540 | 0.033 | −2.765 | 0.006 | 2.220 | 0.137 | −0.746 | 0.456 | 2.635 | 0.105 | −0.811 | 0.418 |
| Emotional | 0.838 | 0.360 | 1.219 | 0.223 | 6.249 | 0.013 | −1.768 | 0.078 | 18.163 | 0.000 | −7.019 | 0.000 | 10.386 | 0.001 | −3.139 | 0.002 | 0.996 | 0.319 | −0.207 | 0.836 |
| **Ordinary people positioning (OP)_photo post** | | | | | | | | | | | | | | | | | | | | |
| Likes | 12.884 | 0.000 | −3.545 | 0.000 | 139.516 | 0.000 | −16.029 | 0.000 | 15.064 | 0.000 | −4.681 | 0.000 | 12.201 | 0.000 | −2.426 | 0.015 | 66.710 | 0.000 | −13.823 | 0.000 |
| Comments | 0.246 | 0.620 | −0.365 | 0.715 | 54.788 | 0.000 | −7.210 | 0.000 | 1.732 | 0.188 | −1.363 | 0.173 | 3.448 | 0.063 | −1.267 | 0.205 | 41.234 | 0.000 | −9.227 | 0.000 |
| Shares | 7.007 | 0.008 | 1.475 | 0.140 | 0.749 | 0.387 | −1.060 | 0.289 | 0.331 | 0.565 | 0.706 | 0.480 | 0.072 | 0.788 | 0.576 | 0.565 | 2.399 | 0.121 | −1.075 | 0.283 |
| Behavioral | 9.248 | 0.002 | −3.091 | 0.002 | 119.507 | 0.000 | −14.976 | 0.000 | 12.457 | 0.000 | −4.312 | 0.000 | 10.499 | 0.001 | −2.220 | 0.027 | 59.181 | 0.000 | −13.219 | 0.000 |
| Love | 52.633 | 0.000 | −5.734 | 0.000 | 0.013 | 0.909 | 1.293 | 0.196 | 1.203 | 0.273 | 1.021 | 0.307 | 49.384 | 0.000 | −8.808 | 0.000 | 25.713 | 0.000 | −3.979 | 0.000 |
| Haha | 34.418 | 0.000 | −4.457 | 0.000 | 3.236 | 0.072 | −1.003 | 0.316 | 0.000 | 0.985 | 0.001 | 0.999 | 16.171 | 0.000 | −6.015 | 0.000 | 0.937 | 0.333 | −0.483 | 0.629 |
| Wow | 16.537 | 0.000 | −2.725 | 0.006 | 8.608 | 0.003 | −1.718 | 0.086 | 0.027 | 0.869 | −0.043 | 0.965 | 45.298 | 0.000 | −9.744 | 0.000 | 6.443 | 0.011 | −1.781 | 0.075 |
| Sorry | 3.334 | 0.068 | −0.948 | 0.343 | 16.954 | 0.000 | −2.741 | 0.006 | 1.208 | 0.272 | 0.551 | 0.582 | 11.862 | 0.001 | −4.972 | 0.000 | 0.294 | 0.587 | 0.278 | 0.781 |
| Anger | 8.384 | 0.004 | −2.143 | 0.032 | 8.520 | 0.004 | −2.023 | 0.043 | 2.405 | 0.121 | −0.779 | 0.436 | 2.703 | 0.100 | −0.829 | 0.407 | 2.429 | 0.119 | −0.786 | 0.432 |
| Emotional | 52.740 | 0.000 | −5.860 | 0.000 | 10.279 | 0.001 | −1.053 | 0.292 | 0.122 | 0.727 | 0.340 | 0.734 | 44.860 | 0.000 | −10.901 | 0.000 | 11.364 | 0.001 | −2.175 | 0.030 |

**Table A4.** Linear regression coefficient of determination and beta (link post and photo post).

| | R | R² | adj. R² | ΔR² | ΔF | df1 | df2 | Sig. F Change | B | Beta | t-Value | p-Value |
|---|---|---|---|---|---|---|---|---|---|---|---|---|
| **Artist positioning (AP)_link post** | | | | | | | | | | | | |
| Behavioral participation | 0.017 [a] | 0.000 | −0.001 | 0.000 | 0.277 | 1 | 908 | 0.599 | −1.009 | −0.017 | −0.526 | 0.599 |
| Emotional participation | 0.011 [a] | 0.000 | −0.001 | 0.000 | 0.105 | 1 | 908 | 0.746 | −0.021 | −0.011 | −0.325 | 0.746 |
| **Artist positioning (AP)_photo post** | | | | | | | | | | | | |
| Behavioral participation | 0.093 [a] | 0.009 | 0.008 | 0.009 | 35.496 | 1 | 4050 | 0.000 | −12.632 | −0.093 | −5.958 | 0.000 |
| Emotional participation | 0.015 [a] | 0.000 | 0.000 | 0.000 | 0.950 | 1 | 4050 | 0.330 | −0.100 | −0.015 | −0.974 | 0.330 |
| **Ordinary people positioning (OP)_link post** | | | | | | | | | | | | |
| Behavioral participation | 0.092 [a] | 0.008 | 0.007 | 0.008 | 6.240 | 1 | 736 | 0.013 | −51.771 | −0.092 | −2.498 | 0.013 |
| Emotional participation | 0.064 [a] | 0.004 | 0.003 | 0.004 | 3.009 | 1 | 736 | 0.083 | −0.904 | −0.064 | −10.735 | 0.083 |

**Table A4.** *Cont*.

|  | R | R² | adj. R² | ΔR² | ΔF | df1 | df2 | Sig. F Change | B | Beta | t-Value | *p*-Value |
|---|---|---|---|---|---|---|---|---|---|---|---|---|
| **Ordinary people positioning (OP)_photo post** | | | | | | | | | | | | |
| Behavioral participation | 0.135 [a] | 0.018 | 0.018 | 0.018 | 102.201 | 1 | 5474 | 0.000 | −1000.178 | −0.135 | −10.109 | 0.000 |
| Emotional participation | 0.052 [a] | 0.003 | 0.003 | 0.003 | 15.036 | 1 | 5474 | 0.000 | −0.519 | −0.052 | −30.878 | 0.000 |

[a] Predicted value: (constant), concert.

**Table A5.** Linear regression coefficient of determination and beta (artist positioning (AP)).

|  | R | R² | adj. R² | ΔR² | ΔF | df1 | df2 | Sig. F Change | B | Beta | t-Value | *p*-Value |
|---|---|---|---|---|---|---|---|---|---|---|---|---|
| **Artist positioning (AP) abstract implication** | | | | | | | | | | | | |
| Huashan1914CreativePark | | | | | | | | | | | | |
| Behavioral participation | 0.043 [a] | 0.002 | 0.002 | 0.002 | 7.463 | 1 | 4050 | 0.006 | −17.549 | −0.043 | −2.732 | 0.006 |
| Emotional participation | 0.010 [a] | 0.000 | 0.000 | 0.000 | 0.397 | 1 | 4050 | 0.529 | 0.194 | 0.010 | 0.630 | 0.529 |
| Design | | | | | | | | | | | | |
| Behavioral participation | 0.091 [a] | 0.008 | 0.008 | 0.008 | 33.647 | 1 | 4048 | 0.000 | −79.372 | −0.091 | −5.801 | 0.000 |
| Emotional participation | 0.048 [a] | 0.002 | 0.002 | 0.002 | 9.218 | 1 | 4048 | 0.002 | −1.999 | −0.048 | −3.036 | 0.002 |
| Art | | | | | | | | | | | | |
| Behavioral participation | 0.031 [a] | 0.001 | 0.001 | 0.001 | 3.863 | 1 | 4050 | 0.049 | −31.333 | −0.031 | −1.965 | 0.049 |
| Emotional participation | 0.009 [a] | 0.000 | 0.000 | 0.000 | 0.346 | 1 | 4050 | 0.556 | −0.450 | −0.009 | −0.588 | 0.556 |
| **Artist positioning (AP) concrete implication** | | | | | | | | | | | | |
| Product | | | | | | | | | | | | |
| Behavioral participation | 0.057 [a] | 0.003 | 0.003 | 0.003 | 12.971 | 1 | 4050 | 0.000 | −66.015 | −0.057 | −3.601 | 0.000 |
| Emotional participation | 0.058 [a] | 0.003 | 0.003 | 0.003 | 13.535 | 1 | 4050 | 0.000 | −3.234 | −0.058 | −3.679 | 0.000 |
| Graphic | | | | | | | | | | | | |
| Behavioral participation | 0.035 [a] | 0.001 | 0.001 | 0.001 | 4.856 | 1 | 4050 | 0.028 | −44.641 | −0.035 | −2.204 | 0.028 |
| Emotional participation | 0.033 [a] | 0.001 | 0.001 | 0.001 | 4.344 | 1 | 4050 | 0.037 | −2.025 | −0.033 | −2.084 | 0.037 |
| House | | | | | | | | | | | | |
| Behavioral participation | 0.048 [a] | 0.002 | 0.002 | 0.002 | 9.534 | 1 | 4050 | 0.002 | 51.256 | 0.048 | 3.088 | 0.002 |
| Emotional participation | 0.052 [a] | 0.003 | 0.002 | 0.003 | 10.851 | 1 | 4050 | 0.001 | 2.622 | 0.052 | 3.294 | 0.001 |

[a] Predicted value: (constant), concert.

**Table A6.** Linear regression coefficient of determination and beta (ordinary people positioning (OP)).

| | R | R² | adj. R² | ΔR² | ΔF | df1 | df2 | Sig. F Change | B | Beta | t-Value | *p*-Value |
|---|---|---|---|---|---|---|---|---|---|---|---|---|
| **Ordinary people positioning (OP) abstract implication** | | | | | | | | | | | | |
| Pier2artCenter | | | | | | | | | | | | |
| Behavioral participation | 0.038 [a] | 0.001 | 0.001 | 0.001 | 7.854 | 1 | 5474 | 0.005 | −71.410 | −0.038 | −2.803 | 0.005 |
| Emotional participation | 0.087 [a] | 0.008 | 0.007 | 0.008 | 42.171 | 1 | 5474 | 0.000 | 2.211 | 0.087 | 6.494 | 0.000 |
| Design | | | | | | | | | | | | |
| Behavioral participation | 0.098 [a] | 0.010 | 0.009 | 0.010 | 52.597 | 1 | 5474 | 0.000 | −448.701 | −0.098 | −7.252 | 0.000 |
| Emotional participation | 0.028 [a] | 0.001 | 0.001 | 0.001 | 4.193 | 1 | 5474 | 0.041 | −1.705 | −0.028 | −2.048 | 0.041 |
| Art | | | | | | | | | | | | |
| Behavioral participation | 0.055 [a] | 0.003 | 0.003 | 0.003 | 16.793 | 1 | 5474 | 0.000 | −179.630 | −0.055 | −4.098 | 0.000 |
| Emotional participation | 0.007 [a] | 0.000 | 0.000 | 0.000 | 0.275 | 1 | 5474 | 0.600 | 0.309 | 0.007 | 0.525 | 0.600 |
| **Ordinary people positioning (OP) concrete implication** | | | | | | | | | | | | |
| Product | | | | | | | | | | | | |
| Behavioral participation | 0.085 [a] | 0.007 | 0.007 | 0.007 | 39.478 | 1 | 5474 | 0.000 | −516.357 | −0.085 | −6.283 | 0.000 |
| Emotional participation | 0.028 [a] | 0.001 | 0.001 | 0.001 | 4.199 | 1 | 5474 | 0.041 | −2.264 | −0.028 | −2.049 | 0.041 |
| Center | | | | | | | | | | | | |
| Behavioral participation | 0.038 [a] | 0.001 | 0.001 | 0.001 | 7.719 | 1 | 5474 | 0.005 | −70.670 | −0.038 | −2.778 | 0.005 |
| Emotional participation | 0.088 [a] | 0.008 | 0.008 | 0.008 | 42.415 | 1 | 5474 | 0.000 | 2.213 | 0.088 | 6.513 | 0.000 |
| Graphic | | | | | | | | | | | | |
| Behavioral participation | 0.075 [a] | 0.006 | 0.005 | 0.006 | 31.258 | 1 | 5474 | 0.000 | −473.516 | −0.075 | −5.591 | 0.000 |
| Emotional participation | 0.019 [a] | 0.000 | 0.000 | 0.000 | 1.978 | 1 | 5474 | 0.160 | −1.601 | −0.019 | −1.407 | 0.160 |

[a] Predicted value: (constant), concert.

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
