# Peer review of "Information Clues and Emotional Intentions: A Case Study of the Regional Image of the Cultural and Creative Community"

_admsci, doi:10.3390/admsci9020039_

Round 1

Reviewer 1 Report

Dear Authors,

The article deals with a very important issue related to information clues and emotional intentions. I would like to outline to you some aspects, some of them need clarification.

In Abstract, the main purpose of the article should be clearly defined as well as the research methodology should be described in more detail.

The introduction should be changed by indicating in turn the following issues: justifying the choice of topic, the main goal of the article, the importance of this topic for the profile of the journal

The way of quoting should be changed and adapted to the requirements of the journal.

Besides, I have general questions:

Why Cloud Gate Dance Theater and Green-in-hand were chosen?

Why was such time range selected?

Were similar tests carried out?

Could the length of the study period affect the results?

Author Response

Point-by-point responses are in PDF file.

Reviewer 2 Report

The research is significant especially for practitioners as it presents a relationship between information clues and emotional intentions, which can be used to get an advantage in the social dialogue. 

Despite the merits of the research, there are some suggestions that can improve the paper: the abstract and manuscript could be improved by clearly stated problem, providing more structured aim, scope and background, stating the principal objectives and scope of the investigation, and summarizing the results, stating the principal conclusions and scientific impact.

Author Response

(The authors gave the same response as above.)
